# Is Your Pregnancy Unwanted or Unhappy? Psychological Correlates of a Cluster of Pregnant Women Who Need Professional Care

**DOI:** 10.3390/healthcare11152196

**Published:** 2023-08-03

**Authors:** Yukiko Ohashi, Mizuki Takegata, Satoru Takeda, Ayako Hada, Yuriko Usui, Toshinori Kitamura

**Affiliations:** 1Faculty of Nursing, Josai International University, Togane 283-8555, Japan; y-ohashi@jiu.ac.jp; 2Kitamura Institute of Mental Health Tokyo, Tokyo 151-0063, Japan; takegata@institute-of-mental-health.jp (M.T.); hada@institute-of-mental-health.jp (A.H.); 3Department of Obstetrics & Gynecology, Faculty of Medicine, Juntendo University, Tokyo 113-8421, Japan; stakeda@juntendo.ac.jp; 4Aiiku Research Institute for Maternal, Child Health and Welfare, Imperial Gift Foundation Boshi-Aiiku-Kai, Tokyo 106-8580, Japan; 5Kitamura KOKORO Clinic Mental Health, Tokyo 151-0063, Japan; 6Department of Community Mental Health & Law, National Institute of Mental Health, National Center of Neurology and Psychiatry, Tokyo 187-8553, Japan; 7Department of Midwifery and Women’s Health, Division of Health Sciences and Nursing, Graduate School of Medicine, The University of Tokyo, Tokyo 113-0033, Japan; yusui@g.ecc.u-tokyo.ac.jp; 8T. and F. Kitamura Foundation for Studies and Skill Advancement in Mental Health, Tokyo 151-0063, Japan; 9Department of Psychiatry, Graduate School of Medicine, Nagoya University, Nagoya 464-8601, Japan

**Keywords:** unintended pregnancy, unhappy pregnancy, cluster analysis

## Abstract

Background: A negative response towards a current pregnancy consists of two elements: unwantedness of and unhappiness about the current pregnancy. Little is clear about whether pregnant women can be categorized in terms of unwantedness and unhappiness as well as what the correlates are of these categories. Methods: An internet survey of 696 women in their first trimester of pregnancy examined the participants’ intention of and emotional reaction towards pregnancy, borderline personality traits, adult attachment style, depression, fear of childbirth, obsessive-compulsive symptoms, fetal bonding disorder, avoidance of taking part in child care, and consideration about termination of pregnancy (TOP). About one third of the participants were followed up with in their second trimester. Results: Two-step cluster analysis using the participants’ intention of and emotional reaction towards pregnancy revealed three groups of pregnant women: those who wanted and were happy about the pregnancy (Cluster 1), those who were unhappy about the pregnancy (Cluster 2), and those who did not intend to be pregnant but were happy about pregnancy (Cluster 3). Cluster 2 women, but not Cluster 3 women, were likely to be single, with borderline personality traits as well as unstable adult attachment styles, accompanied by depression, fear of childbirth, obsessive-compulsive symptoms, and fetal bonding disorder. They were more likely to avoid caring for the baby after childbirth and consider TOP. Conclusion: Expectant women who were unhappy about their pregnancy were at risk of psychological adjustment and need specific perinatal mental health assessment and care.

## 1. Introduction

Pregnancy is not always a joyful event. It is often accompanied by psychological maladjustments such as depression [1,2], obsessive compulsive disorder [3,4,5], tokophobia [6,7], or fetal bonding disorder [8] to name just a few. One of the predictors of psychological maladjustment during pregnancy is a negative response towards the current pregnancy [9,10]. However, a negative response towards pregnancy may have two facets. One is the unwantedness (unintendedness) of the current pregnancy. The pregnancy was not planned, expected, or even avoided. Another is unhappiness about the current pregnancy. A woman feels unhappy or even disappointed by knowing that she has become pregnant.

Previous research, however, did not necessarily pay careful attention towards precise categorization and different effects of unwantedness of and unhappiness towards a current pregnancy. For example, among women who gave birth via Caesarean section, unwanted pregnancy was linked to neonatal death (causes not specified) within 28 days after childbirth [11]. A study in Uganda reported association between unwanted pregnancy and postpartum depression after childbirth [12] and dissatisfaction during pregnancy [13]. Unplanned pregnancy was reported to be associated with antenatal depression [14]. There is another study that showed that unintended pregnancy ushered postnatal depression [15]. These studies, however, did not examine unhappiness about a pregnancy. In a large epidemiological study of pregnant women, cases of antenatal depression were significantly more likely to claim both ‘negative emotional response’ towards and unwantedness of the current pregnancy [2]. However, in a follow-up study of pregnant women until one month after childbirth, women with unintended pregnancies were twice as likely to have postnatal depression at 12 months after childbirth [16]. This study, however, did not assess the degree of happiness about the pregnancy. Ohashi et al. [17] reported that maternal infant bonding disorder was predicted by a ‘negative emotional response’ towards a pregnancy but not by unwantedness of a pregnancy. The fact that these two categories (unwantedness and unhappiness) do not necessarily coexist (hence are not interchangeable) and indeed had different psychological impacts was noted by the study of Sable and Libbus [18]. They studied women whose pregnancies were unintended. Nevertheless, about half of them reported that they were somewhat or very happy about their pregnancy. The proportion of women who were uncertain about the future of their pregnancy (possible termination of pregnancy) and who considered adoption was highest among those who were unhappy about the pregnancy. They emphasized the importance of differentiating between pregnancy intention (wantedness) and happiness. In another study, compared with women who felt very happy to be pregnant, the rate of postnatal depression (Edinburgh Postnatal Depression Scale was scored 9 or higher) among women who felt their pregnancy was unintended but happy, those who felt their pregnancy was unintended and confused, and women who felt troubled was 1.25, 1.64, and 2.25 times higher, respectively [19]. This is in line with a recent report from Japan [20]. Hence, women’s emotional reactions towards their pregnancy (i.e., happy or unhappy) is no less important than a plan for pregnancy in terms of effects upon psychological adjustment.

If unwantedness of and unhappiness towards a current pregnancy are not necessarily the same, it may be feasible to speculate that pregnant women will be categorized by patterns of the two and will be correlated with demographic, obstetric, and psychological features differently. This difference, if proved, will provide clinical implications for perinatal health care.

This study focused on three points. First, do pregnant women consist of discrete categories (clusters) in terms of unwantedness of and unhappiness about the current pregnancy? Second, what are the possible causal correlates of the categories of pregnant women such as personality and adult attachment? Third, what are the possible consequences of the categories of pregnant women such as depression, obsessive-compulsive symptoms, and fetal bonding as well as the consequences (outcomes) of a pregnancy?

## 2. Methods

### 2.1. Study Procedure and Participants

This is a secondary analysis using data already reported. This focused the psychological health issues of expectant women in Japan during the second wave of the COVID-19 pandemic [21]. Briefly, this was an internet survey of which focus was a pregnant woman population at 12 to 15 weeks’ gestational age (Time 1: T1). The T1 survey was conducted during a two-week period (from the 7th to 21st of December 2020). The Internet applications we used in this study was Luna Luna and Luna Luna Baby (MTI Ltd., Tokyo, Japan). Participant expectant women were 696. We applied no specific exclusion criteria except for a lack of command of Japanese reading. We assured anonymity of the participants and participation was totally voluntary. The main part of the questionnaire was ushered by an information page including the aims of the study, affiliations, information about informed consent, and the address of the consultation desk for the research.

The second invitation to the net survey was announced to the same women. This follow-up survey (Time 2 [T2]) was conducted about 10 weeks later in March 2021. A purpose of this follow-up survey was to examine which participants did not respond in T2 (attrition) (see below). We used the same set of questionnaires as used at T1, except for basic demographic and obstetric variables (which were not repeated at T2). There were no missing values because this survey set the items as forced questions.

### 2.2. Measurements

Reaction to the current pregnancy: Both unwanted pregnancy and unhappy pregnancy were assessed by a single ad hoc question each rated with a 5-point scale. The question for unwanted pregnancy was “Did you want to be pregnant?” with ratings of “I did not want be pregnant” = 0, “I wanted but it was bit earlier” = 1, “I left it to take its own course” = 2, “I wanted it” = 3, and “I wanted and tried to have a baby” = 4. The question for unhappy pregnancy was “How did you feel when you knew you got pregnant?” with ratings of “very displeased” = 0, “relatively displeased” = 1, “neither pleased nor displeased” = 2, “relatively pleased” = 3, and “very pleased” = 4.

Demographic and obstetric variables: They included (a) the participant’s age, (b) the number of past pregnancies, (c) the number of past deliveries, and (d) the marital status (single/married).

Borderline personality traits: We used the short version (IPO-SV) [22] of the Personality Organization Inventory (IPO) [23] to assess borderline personality traits. The IPO-SV has nine items on a 7-point scale containing three subscales (primitive defense [PD], identity diffusion [ID], and reality testing [RT] disturbance.

Adult attachment style: The participants’ attachment towards their partners was rated by the Japanese version [24] of the Relationship Questionnaire (RQ) [25]. This has only four items each describing different styles of adult attachment: Secure, Fearful, Preoccupied, and Dismissing. Each item was rated with a 7-point scale (Does not apply to me at all = 0 to Applies to me very much = 6). The total score was derived from addition of the scores of Fearful, Preoccupied, and Dismissing, which was then subtracted by the score of Secure. A higher score of the RQ indicates a higher insecurity of attachment style.

Depression: The first two items of a Major Depressive Episode (MDE) for the last 2 weeks—depressed mood and lack of interest—were asked on a 4-point scale: 0 = none, 1 = a few days, 2 = more than half the duration, and 3 = almost every day. The set of these two items was used to detect a MDE because research showed that it would predict a MDE with reasonable accuracy [26,27,28,29,30,31,32,33]. As a rough indicator of a MDE, we defined a participant having a MDE if either or both of the two items was rated as present almost every day for the previous 2 weeks.

Fear of childbirth (FOC): In order to assess FOC, we used the Japanese version A [34] of the Wijma Delivery Expectancy/Experience Questionnaire (WDEQ) [35]. We used the WDEQ version A consisting of 33 items each with a 5-point scale. Higher scores of the scale indicate more severe FOC. Unfortunately, Item 31 was erroneously deleted in the present study.

Obsessive-compulsive symptoms: To measure obsessive and compulsive symptoms, we used the Japanese version [36] of the Obsessive-Compulsive Inventory-Revised (OCI-R) [37]. This has 18 items with a 5-point scale. In order to match with the other scale rating model as well as to increase internal consistency, we changed the grading from a 5-point to a 7-point scale.

Foetal bonding disorder: In order to assess the participant’s emotions towards the fetus, we used the abridged version of the Scale of the Parent to Baby Emotion (SPBE) [38]. This consists of 6 basic and 4 self-conscious emotion subscales each rated by two items with a 7-point scale. The basic emotions include Happiness, Anger, Fear, Sadness, Disgust, and Surprise [39,40] whereas the self-conscious emotions include Shame, Guilt, and Alpha- and Beta-prides [41,42]. They were preceded by the instruction “How strongly did you feel these emotions when you imagined your baby in your womb?” Two subscale scores were calculated by adding item scores: positive fetal bonding based on Happiness and Alpha- and Beta-prides; and negative fetal bonding based on Anger, Fear, Sadness, Disgust, Surprise, Shame, and Guilt.

Avoidance of taking part in child care: We used an ad hoc question asking “Do you plan to look after your baby after childbirth?” with a 7-point Likert type scale from “not at all” to “very much so” (reversed).

Consideration of termination of pregnancy (TOP): An ad hoc question was used to assess attitude towards their current pregnancy: “Are you considering terminating your pregnancy?” with a 7-point Likert-type scale from “not at all” to “very much so.” Higher scores indicate a desire for TOP.

### 2.3. Data Analysis

After calculating mean, SD, skewness, and kurtosis of all of the variables, we correlated scores of unwanted pregnancy and unhappy pregnancy. A two-step cluster analysis was performed using the scores of unintended and unhappy pregnancies with log likelihood method and Bayesian Information Criterion (BIC) to define the number of clusters. Cluster analysis identifies types according to individual differences [43]. There are a few techniques of cluster analysis. Agglomerative hierarchical cluster analysis that is widely used is, however, characterized by ambiguity of determining the appropriate number of clusters. Another population method, K-means cluster analysis [44], demands the researcher to determine the number of clusters beforehand. We used two-step cluster analysis in this study. The procedure starts with the construction of a cluster features tree that creates ‘nodes’ containing multiple cases. In the second step, agglomerative clustering is used to produce a range of solutions. It automatically confirms the possible maximum number of clusters. This will be followed by determination of the best cluster model in terms of the highest distance increase (measured by Schwarz’s Bayesian Criterion [45] or Akaike Information Criterion [46]) between the two closest cluster models during each stage of the hierarchical clustering [47,48]. Two-step cluster analysis can also deal with large data files efficiently. Two-step cluster analysis has been used in many psychosocial studies (e.g., [49,50,51]). As compared with the other methods that identify groups (clusters) of subjects including latent class analysis, the performance of two-step analysis is equally excellent. Silhouette coefficient analysis was taken to evaluate the silhouette of clusters [52,53]. Little has been described as to the sufficient number of cases needed for cluster analysis. A rule of thumb usually suggested is 100, 100–200, and 200 or more as ‘small’, ‘medium’, and ‘large’, respectively. Our sample size was substantially larger than required by this rule. The clusters were compared in terms of the scores of unintended and unhappy pregnancies. To determine the construct validity of the obtained clusters, we compared the clusters’ mean scores of demographic and obstetrics features, borderline personality traits, adult attachment style, perceived impact of pregnancy, depression, fear of childbirth, obsessive-compulsive symptoms, and fetal bonding disorder.

### 2.4. Ethical Considerations

Approval was given by the Institutional Review Board (IRB) of the Kitamura Institute of Mental Health Tokyo (No. 2020101501). All participants provided electronic informed consent after reading and understanding the study’s rationale and procedure. All procedures associated with this study comply with the ethical standards of the National and Institutional Committees on Human Experimentation and with the Helsinki Declaration of 1975 as revised in 2008.

## 3. Results

About half (40%) of the participants reported that they wanted and tried to have a baby and two thirds (66%) at least wanted a baby (Table 1). About one in ten participants (11%) reported that they did not want to be pregnant at least at that time point. In discussing whether they were happy about the current pregnancy, 80% of the women said they were “very pleased” and 95% of them said they were at least pleased. The unwanted pregnancies and unhappy pregnancies were significantly correlated: *r* = 0.44 (*p* < 0.001).

Two-step cluster analysis yielded three clusters with the silhouette coefficient = 0.6 (Table 2). Cluster 1 consisted of more than half of the participants (*n* = 421) and was characterized by happiness and intendedness of pregnancy (Table 2). Cluster 2 (*n* = 139) was characterized by unhappiness of pregnancy. On the other hand, Cluster 3 (*n* = 136) was characterized by unintendedness but happiness of pregnancy. We, therefore, named the three Clusters Wanted and Happy, Unhappy, and Unwanted but Happy, respectively. By spotting each case of the three clusters in Table 1, it was found that the Cluster 1 (Wanted and Happy) cases were located in the right two cells of the ‘very pleased’ row (Table 1 yellow cells) whereas the Cluster 3 (Unwanted but happy) cases were located in the left three cell of the ‘very pleased’ row (Table 1, blue cells). The Cluster 2 (Unhappy) cases were located in the remaining cells.

Age was significantly but slightly higher in Cluster 1. Cluster 1 was also characterized by a smaller number of past deliveries. The three clusters did not differ in terms of gestation week or number of past pregnancies. Cluster 2 had a higher rate of single marital status (5% vs. 0%).

As compared to Clusters 1 and 3, Cluster 2 was characterised by higher borderline personality traits, lower Self- and Other-models, higher depression, fear of childbirth, and obsessive-compulsive symptoms. Cluster 2 was also characterised by low positive and high negative foetal bonding scores.

With respect to how to cope with the pregnancy and rearing after chirldbirth, as compared to Clusters 1 and 3, Cluster 2 was characterised by higher avoidance of taking part in child care and a plan to terminate the pregnancy. Because we did not know the course of the participant women regarding continuing the pregnancy, we studied the attrition of participants at Time 2 that was about 10 weeks later as an alternative measure. About one third (151/421) of the women answered the T2 questionnaire. The attrition rate did not differ between the Clusters (Cluster 1 = 64%, Cluster 2 = 65%, and Cluster 3 = 67%). The attrition of the Cluster 2 women was not predicted by the degree of avoidance of taking part in child care or a plan to terminate the pregnancy.

## 4. Discussion

This study demonstrated that although unwantedness of and unhappiness about pregnancy were correlated with one another to some extent, there were three groups of pregnant women based on the patterns of these two variables: those who wanted and were happy about their pregnancy (Cluster 1), those who were unhappy about their pregnancy (Cluster 2), and those who did not intend to be pregnant but were happy about their pregnancy (Cluster 3). These groups did not differ in terms of gestation week and the number of past pregnancies. However, women in these groups differed in several areas. It is, however, of note that the naming of Clusters may be simplistic. Thus, “Wanted and Happy” and “Unwanted but Happy” women all answered “very pleased” whereas “Unhappy” women varied between “very unpleased” to “relatively unpleased” (but never “very pleased”).

All of the single women belonged to Cluster 2 but they were less than 10% of the Cluster 2 women. Hence, one reason for an unhappy pregnancy is an unmarried status, but it only explains a small portion of the feelings. The women in Cluster 2 scored higher in borderline personality traits and poorer self and other models of adult attachment style. These scores suggested that Cluster 2 pregnant women might be characterized by fragile personal and interpersonal maturation. In addition, the Cluster 2 women, but not the Cluster 3 women, were characterized by higher scores of depression, fear of childbirth, obsessive-compulsive symptoms, and fetal bonding disorders. Women with unhappy pregnancies are at high risk of psychological maladjustment during pregnancy. Therefore, they need specific perinatal mental health support and care. Focusing clinical attention only on depression is too simplistic. It should cover a wider area of antenatal psychopathology.

More consideration should be paid to the characteristics of Cluster 3 women. Why did they feel happy about the signs of pregnancy although they did not want or intend to be pregnant? They may have encountered a positive response from their partner. Their social circumstances may have made them more accepting towards a pregnancy. It may be that Cluster 2 women may be less likely to seek support from others. For example, only about 10% of women with postnatal depression seek professional support [54]. Jones [55] listed barriers for women with perinatal depression to seeking help including social (e.g., stigma), structural (e.g., provider unavailable), and instrumental (e.g., cost) factors. Similarly, women who feel unhappy about their pregnancy may be less likely to seek support because of feeling shame that they do not feel happy about their pregnancy.

Again, Cluster 2 women but not Cluster 3 women were found to be unmotivated to care for the baby after childbirth and to consider termination of their pregnancy. The outcome of the pregnancy among these women was of clinical concern. We were unable to follow the participants until delivery but we managed to follow about one third of them until the second trimester. It is not that the Cluster 2 women were likely to retire from the follow-up study or the desire to terminate the pregnancy predicted attrition. Thus, Cluster 2 women remained in the T2 study and were still considering terminating their pregnancy. Considering the impact of maternal perinatal mental disorders on different aspects of the fetus and child [56], expectant women characterized by unhappiness about their pregnancy should be screened and provided with mental health care with special attention to consequent mother-to-fetus and mother-to-infant relationships. Unhappiness about a pregnancy might be one of the earliest/easiest screening indicators of undesirable maternal and infant mental health consequences.

It is of note that our study was conducted during the pandemic of COVID-19 in Japan. The stress related to the restrictions for COVID-19 may have affected the study results e.g., [57,58,59]. Impacts of the pandemic on perinatal mental health may vary from a country to another. In Japan, Takubo et al. [60] studied women who had completed a maternity health check-up at a regional hospital during the period from 1 April 2017, to 31 December 2020. The women were divided into four groups (three Before COVID-19 groups and a During COVID-19 group). They found no differences among the four groups in terms of depression (the total score of the Edinburgh Postnatal depression Score [EPDS]) and mother-to-infant bonding disorders. However, the anxiety subscale scores of the EPDS were significantly higher while the anhedonia and depression subscale scores of the EPDS were significantly lower in the During COVID-19 group than the Before COVID-19 group. Our result may be biased by anxiety caused the Pandemic. Further consideration is difficult due to lack of comparison group in our study.

This study is not without limitations. Participation in this web survey was voluntary. Therefore, the results might be biased by participants’ motivation that might have links with the variables studied here. All of the data were derived from a questionnaire distributed vias a single website. We should be cautious about possible confounding by variables we are not aware of. Direct interviews may yield different results. The attrition rate from T1 to T2 was substantial. More rigorous follow-up studies may give us a more accurate picture of the pregnancy outcomes and trajectory of mental states. Finally, we were unaware of how many women were currently or as a past history mentally ill. Lack of information of psychiatric diagnosis is a major limitation of the current study.

To further advance the understanding of the psychological aspects of unwantedness and unhappiness during pregnancy, we may consider expansion of our study. It may include investigating the long-term impacts of unwanted and unhappy pregnancies on maternal and child well-being, exploring potential interventions to support women in Cluster 2, and examining the role of social support in mitigating the negative psychological outcomes associated.

## 5. Conclusions

Taking into consideration these drawbacks, this study showed that expectant women consisted of three groups in terms of unwantedness of and unhappiness about their pregnancy and those who did not feel happy about their pregnancy were likely to be single, with borderline personality traits as well as unstable adult attachment, accompanied by depression, fear of childbirth, obsessive-compulsive symptoms, and fetal bonding disorder. They were more likely to avoid caring for the baby after childbirth and to consider TOP. Therefore, women who feel unhappy about their pregnancy should be a target of specialized intensive mental health care during pregnancy.

## Figures and Tables

**Table 1 healthcare-11-02196-t001:** Distribution of participants regarding unwantedness and unhappiness.

Unhappiness	Unwanted Pregnancy					
	0: Not Want Be Pregnant	1: Bit Earlier	2: Left It to Take Its Own Course	3: Wanted It	4: Wanted and Tried to Have a Baby	Total
0: very unpleased	1	0	0	0	1	2
1: relatively displeased	3	2	1	0	1	7
2: neither pleased nor displeased	6	14	6	1	2	29
3: relatively pleased	8	34	27	15	17	101
4: very pleased	4	49	83	166	255	557
total	22	99	117	182	276	696

**Table 2 healthcare-11-02196-t002:** Two-step cluster analysis of the participants.

	Cluster 1 Wanted and Happy	Cluster 2 Unhappy	Cluster 3 Unwanted but Happy	*F*	Tukey Posthoc Comparison
*n*	421	139	136		
Happiness	4.0 (0.0)	2.7 (0.7)	4.0 (0.0)	71.8 ***	2 < 1, 3
Intendedness	3.6 (0.5)	1.8 (1.3)	1.6 (0.6)	53.3 ***	3 < 2 < 1
Age	32.2 (4.3)	31.4 (5.1)	30.7 (4.6)	6.0 **	3 < 1
Gestation week	13.4 (1.1)	13.4 (1.1)	13.3 (1.1)	0.0	
Number of past pregnancies	0.71 (1.03)	0.91 (1.31)	0.69 (0.99)	1.9	
Number of past deliveries	0.33 (0.63)	0.53 (0.95)	0.36 (0.71)	4.1 *	1 < 2
Marital status (single/married)	0/421	6/133	0/136	χ^2^ = 24.252 ***	
IPO total	13.4 (9.2)	17.1 (10.4)	13.9 (9.7)	8.0 ***	1 = 3 < 2
RQ: Self-model	3.33 (2.96)	2.33 (3.80)	3.76 (2.77)	7.7 ***	2 < 1 = 3
RQ: Other-model	3.89 (2.67)	2.80 (3.55)	3.96 (3.07)	7.5 ***	2 < 1 = 3
Depression	1.01 (1.33)	1.68 (1.75)	1.13 (1.39)	11.5 ***	1 = 3 < 2
Tokophobia: WDEQ	59.6 (19.5)	70.7 (23.8)	57.9 (20.2)	17.9 ***	3 = 1 < 2
Obsessive-compulsive symptoms	27.6 (16.5)	30.8 (18.3)	25.6 (14.9)	3.5 *	3 < 2
Positive foetal bonding	24.8 (4.7)	18.7 (6.4)	24.2 (5.5)	71.8 ***	2 < 1 = 3
Negative foetal bonding	12.5 (10.3)	24.2(14.9)	15.0 (1.2)	53.3 ***	1 = 3 < 2
Avoiding caring for the baby after childbirth	0.74 (1.22)	1.17 (1.53)	0.68 (1.07)	7.1 ***	3 = 1 < 2
Considering TOP	0.05 (0.28)	0.80 (1.39)	0.23 (0.77)	53.7 ***	1 < 3 < 2

* *p* < 0.05, ** *p* < 0.01, *** *p* < 0.001. IPO, Inventory of Personality Organization; RQ, Relationship Questionnaire; TOP, termination of pregnancy; WDEQ, Wijma Delivery Expectancy/Experience Questionnaire.

## Data Availability

The datasets used and analysed in the present study are available from the corresponding author upon reasonable request.

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
