# Peer review of "Is Your Pregnancy Unwanted or Unhappy? Psychological Correlates of a Cluster of Pregnant Women Who Need Professional Care"

_healthcare, 2023, doi:10.3390/healthcare11152196_

Round 1

Reviewer 1 Report (Previous Reviewer 1)

In the statistical methods section, the author states: "A two-step cluster analysis was performed using the scores of unintended and unhappy pregnancies with log likelihood method and Bayesian Information Criterion (BIC) to define the number of clusters. Cluster analysis identifies types according to individual differences [43]." in addition, looking at "Table 1. Distribution of participants regarding unwantedness and unhappiness.", a new statement under Table 1: "The Cluster 2 (Unwanted and Unhappy) cases were located in the reaining cells". I take (reaining=remaining)

My concern is: if Table 1 and the Cluster analysis (creating the 3 clusters is based on similar variables  "unwantedness" and "unhappiness", then why is there a discrepancy between Table 1 and the Cluster 2 (Unwanted and Unhappy) frequencies?  (n=139) can not be matched up in Table 1.  If the clustering has been performed correctly on the same variable as used in Table 1, should the 3 clusters' frequencies and frequencies in Table 1 be consistent? 

Author Response

In the statistical methods section, the author states: "A two-step cluster analysis was performed using the scores of unintended and unhappy pregnancies with log likelihood method and Bayesian Information Criterion (BIC) to define the number of clusters. Cluster analysis identifies types according to individual differences [43]." in addition, looking at "Table 1. Distribution of participants regarding unwantedness and unhappiness.", a new statement under Table 1: "The Cluster 2 (Unwanted and Unhappy) cases were located in the reaining cells". I take (reaining=remaining)

Thank you for your comments. We revised the word to "remaining".

My concern is: if Table 1 and the Cluster analysis (creating the 3 clusters is based on similar variables  "unwantedness" and "unhappiness", then why is there a discrepancy between Table 1 and the Cluster 2 (Unwanted and Unhappy) frequencies?  (n=139) can not be matched up in Table 1.  If the clustering has been performed correctly on the same variable as used in Table 1, should the 3 clusters' frequencies and frequencies in Table 1 be consistent?

We agree. We renamed Cluster 2 as “Unhappy”. In addition, we hope the following sentences added to Result will answer the reviewer’s enquiry.

By spotting each case of the three clusters in Table 1, it was found that the Clutser 1 (Wanted and Happy) cases were located in the right two cells of the ‘very pleased’ row whereas the Cluster 3 (Unwanted but happy) cases were located in the left three cell of the ‘very pleased’ row. The Cluster 2 (Unwanted and Unhappy) cases were located in the remaining cells.

Reviewer 2 Report (New Reviewer)

I have reviewed the manuscript "Is your pregnancy unwanted or unhappy? Psychological correlates of a cluster of pregnant women who need professional care" and I appreciate the authors' efforts in this study. However, I have identified several areas to address before the manuscript can be considered for publication.

Specifically, the Authors should consider the following recommendations:

-       The Methods section should be revised to provide more information on the study's procedures and sample characteristics. For instance, the authors should provide more information on the psychometric characteristics of the questionnaires used in the study. Additionally, the authors should provide a rationale for the sample size and ensure that the sample is appropriately characterized to generalize the results.

-       Since the study was conducted during the pandemic, it is important to highlight how the stress related to the restrictions for COVID-19 may have affected the study results. In this regard, it would be interesting to also briefly mention the impact of the pandemic on the psychological well-being of pregnant and postpartum women by citing some recent articles on the topic: DOI: 10.1515/jpm-2021-0368; DOI: 10.3390/ijerph19042219; DOI: 10.3390/ijerph17165933.

-       The Authors should provide more detailed information on the study's limitations, including the potential biases and confounding factors that may have influenced the results.

-       To further advance the understanding of the psychological aspects of unwantedness and unhappiness during pregnancy, it would be beneficial to suggest directions for future research. This could include investigating the long-term impacts of unwanted and unhappy pregnancies on maternal and child well-being, exploring potential interventions to support women in Cluster 2, and examining the role of social support in mitigating the negative psychological outcomes associated.

I recommend a linguistic revision of the manuscript before resubmitting it to improve its readability.

Author Response

The Methods section should be revised to provide more information on the study's procedures and sample characteristics. For instance, the authors should provide more information on the psychometric characteristics of the questionnaires used in the study. 

We guess that the reviewer wanted us to report psychometric properties such as Cronbach’s alpha coefficient, However, because each measure only consists of a few items, Cronbach’s alpha is expected to be lower than it should be. In addition, we added references about previous reports about details of each measure’s reliability and validity.

Additionally, the authors should provide a rationale for the sample size and ensure that the sample is appropriately characterized to generalize the results.

We added the following sentences.

Little has been described as to the sufficient number of cases needed for cluster analysis. A rule of thumb usually suggested is 100, 100 to 200, and 200 or more as ‘small’, ‘medium’, and ‘large’, respectively. Our sample size was substantially larger than required by this rule.

Since the study was conducted during the pandemic, it is important to highlight how the stress related to the restrictions for COVID-19 may have affected the study results. In this regard, it would be interesting to also briefly mention the impact of the pandemic on the psychological well-being of pregnant and postpartum women by citing some recent articles on the topic: DOI: 10.1515/jpm-2021-0368; DOI: 10.3390/ijerph19042219; DOI: 10.3390/ijerph17165933.

We added the following paragraph in Discussion.

It is of note that our study was conducted during the pandemic of COVID-19 in Japan. The stress related to the restrictions for COVID-19 may have affected the study results [e.g., 52,53,54]. Impacts of the pandemic on perinatal mental health may vary from a country to another. In Japan, Takubo et al. [55] studied women who had completed a maternity health check-up at a regional hospital during the period from April 1, 2017, to December 31, 2020. The women were divided into four groups (three Before COVID-19 groups and a During COVID-19 group). They found no differences between among the four groups in terms of depression (the total score of the Edinburgh Postnatal depression Score [EPDS]) and mother-to-infant boning disorders. However, the anxiety subscale scores of the EPDS were significantly higher while the anhedonia and depression subscale scores of the EPDS were significantly lower in the During COVID-19 group than the Before COVID-19 group. Our result may be biased by anxiety caused the Pandemic. Further consideration is difficult due to lack of comparison group in our study.

The Authors should provide more detailed information on the study's limitations, including the potential biases and confounding factors that may have influenced the results.

We amended the paragraph in question in Discussion as follows;

This study is not without limitations. Participation in this web survey was voluntary. Therefore, the results might be biased by participants’ motivation that might have links with the variables studied here. All the data were derived from a questionnaire distributed vias a single website. We should be cautious about possible confounding by variables we are not aware of. Direct interviews may yield different results. The attrition rate from T1 to T2 was substantial. More rigorous follow-up studies may give us a more accurate picture of the pregnancy outcomes and trajectory of mental states.

To further advance the understanding of the psychological aspects of unwantedness and unhappiness during pregnancy, it would be beneficial to suggest directions for future research. This could include investigating the long-term impacts of unwanted and unhappy pregnancies on maternal and child well-being, exploring potential interventions to support women in Cluster 2, and examining the role of social support in mitigating the negative psychological outcomes associated.

We agree with and thankful for recommendations of the reviewer. We added the following sentences in Discussion. 

To further advance the understanding of the psychological aspects of unwantedness and unhappiness during pregnancy, we may consider expansion of our study. It may include investigating the long-term impacts of unwanted and unhappy pregnancies on maternal and child well-being, exploring potential interventions to support women in Cluster 2, and examining the role of social support in mitigating the negative psychological outcomes associated.

I recommend a linguistic revision of the manuscript before resubmitting it to improve its readability.

We had a bilingual person to check the text again, 

Reviewer 3 Report (New Reviewer)

This manuscript entitled “Is your pregnancy unwanted or unhappy? Psychological correlates of a cluster of pregnant women who need professional care” deals with a study that classified pregnant women by their emotional reaction toward pregnancy and investigated characteristics of the classified women. The overall paper is written in a concise and readable manner. However, the following several concerns need to be addressed in this manuscript.

1) The question about unwanted pregnancy developed by the authors seems somewhat incongruous. Does "I felt it to take its own course" indicate a more substantial degree of wanting to be pregnant than "I wanted but it was a bit earlier"? To begin with, this seems to indicate a different dimension. It would be worthwhile to consider excluding this response from the analysis.

2) The Method section should describe how the Time 2 data will be analyzed.

3) The names of each cluster are very clearly defined and easy to understand. However, the name of Cluster 2 can be misleading: although the average Intendedness score is low, it includes many of those who answered "wanted it" or "wanted and tried to have a baby."

4) It would be helpful to describe more detailed results for the cluster analysis, which is the most important result of this study. Although the number of clusters was automatically determined, it would be better to present the BIC and AIC according to the number of clusters.

5) The results of the analysis in Time 2 should not only show dropouts but should also show the same items as in Table 2 (Time 1). It would strengthen the discussion on Cluster 2 as currently described.

6) If the authors have investigated how many of the participants in this study had a mental illness, they should present it. This could significantly affect the results and may be a limitation of this study.

Author Response

1) The question about unwanted pregnancy developed by the authors seems somewhat incongruous. Does "I felt it to take its own course" indicate a more substantial degree of wanting to be pregnant than "I wanted but it was a bit earlier"? To begin with, this seems to indicate a different dimension. It would be worthwhile to consider excluding this response from the analysis.

We consider that “I felt it to take its own course” means a direction towards “I wanted it” while “I wanted but it was a bit earlier” means a direction towards “I did not want it at least right now”. We hope the reviewer would be concurrent with our image of expression.

2) The Method section should describe how the Time 2 data will be analyzed.

We inserted the following the next sentence in Study procedure and participants.

A purpose of this follow-up survey was to examine which participants did not respond in T2 (attrition) (see below).

3) The names of each cluster are very clearly defined and easy to understand. However, the name of Cluster 2 can be misleading: although the average Intendedness score is low, it includes many of those who answered "wanted it" or "wanted and tried to have a baby."

The reviewer’s comment is succinct. We renamed Cluster 2 as “Unhappy”. In addition, we added the following sentences in the first paragraph in Discussion.

It is, however, of note that the naming of Clusters may be simplistic. Thus, “Wanted and Happy” and “Unwanted but Happy” women all answered “very pleased” whereas “Unwanted and Unhappy” women varied between “very unpleased” to “relatively unpleased” (but never “very pleased”).

4) It would be helpful to describe more detailed results for the cluster analysis, which is the most important result of this study. Although the number of clusters was automatically determined, it would be better to present the BIC and AIC according to the number of clusters.

Our knowledge does not catch up with reviewer’s comment. We are not aware of presenting the BIC and AIC according to the number of clusters in a two-step cluster method.

5) The results of the analysis in Time 2 should not only show dropouts but should also show the same items as in Table 2 (Time 1). It would strengthen the discussion on Cluster 2 as currently described.

We consider that due to a fairly large attrition rate deeper analyses of T2 data may be misleading. Hence, we used the T2 data for discussion of attrition.

6) If the authors have investigated how many of the participants in this study had a mental illness, they should present it. This could significantly affect the results and may be a limitation of this study.

The reviewer is correct. We added the following sentence in Discussion as a major limitation of the study.

Finally, we were unaware of how many women were currently or as a past history mentally ill. Lack of information of psychiatric diagnosis is a major limitation of the current study.

Round 2

Reviewer 3 Report (New Reviewer)

The authors addressed the comments appropriately. I greatly appreciate the opportunity to review this manuscript.

Author Response

1) The question about unwanted pregnancy developed by the authors seems somewhat incongruous. Does "I felt it to take its own course" indicate a more substantial degree of wanting to be pregnant than "I wanted but it was a bit earlier"? To begin with, this seems to indicate a different dimension. It would be worthwhile to consider excluding this response from the analysis.

We consider that “I felt it to take its own course” means a direction towards “I wanted it” while “I wanted but it was a bit earlier” means a direction towards “I did not want it at least right now”. We hope the reviewer would be concurrent with our image of expression.

2) The Method section should describe how the Time 2 data will be analyzed.

We inserted the following the next sentence in Study procedure and participants.

A purpose of this follow-up survey was to examine which participants did not respond in T2 (attrition) (see below).

3) The names of each cluster are very clearly defined and easy to understand. However, the name of Cluster 2 can be misleading: although the average Intendedness score is low, it includes many of those who answered "wanted it" or "wanted and tried to have a baby."

The reviewer’s comment is succinct. We renamed Cluster 2 as “Unhappy”. In addition, we added the following sentences in the first paragraph in Discussion.

It is, however, of note that the naming of Clusters may be simplistic. Thus, “Wanted and Happy” and “Unwanted but Happy” women all answered “very pleased” whereas “Unwanted and Unhappy” women varied between “very unpleased” to “relatively unpleased” (but never “very pleased”).

4) It would be helpful to describe more detailed results for the cluster analysis, which is the most important result of this study. Although the number of clusters was automatically determined, it would be better to present the BIC and AIC according to the number of clusters.

Our knowledge does not catch up with reviewer’s comment. We are not aware of presenting the BIC and AIC according to the number of clusters in a two-step cluster method.

5) The results of the analysis in Time 2 should not only show dropouts but should also show the same items as in Table 2 (Time 1). It would strengthen the discussion on Cluster 2 as currently described.

We consider that due to a fairly large attrition rate deeper analyses of T2 data may be misleading. Hence, we used the T2 data for discussion of attrition.

6) If the authors have investigated how many of the participants in this study had a mental illness, they should present it. This could significantly affect the results and may be a limitation of this study.

The reviewer is correct. We added the following sentence in Discussion as a major limitation of the study.

Finally, we were unaware of how many women were currently or as a past history mentally ill. Lack of information of psychiatric diagnosis is a major limitation of the current study.

This manuscript is a resubmission of an earlier submission. The following is a list of the peer review reports and author responses from that submission.

Round 1

Reviewer 1 Report

Comments to the Editor and Author:

Dear Editor,

This manuscript with the title of “Is your pregnancy unwanted or unhappy? Psychological correlates of a cluster of pregnant women who need professional care”, is a brief manuscript. Its main aim is to characterise current pregnancies in terms of unwantedness and unhappiness about the pregnancy. The main limitation of this work is not conducting proper statistical analysis and “overpowering” the analysis as reported. I am sorry it might be disappointing for the authors, but I can not recommend the publication of this paper. In support of my decision, I would like to explain as follow:

1.      The author claimed conducting a two-stage cluster analysis, but it is not clear from the statistical methods, how this analysis was conducted.

2.      Table 1, results show that the characterisation may not have been performed properly. Frequencies in the upper left side of table 1 did not convince me that the number (n = 139) for “Cluster 2: unwanted or unhappy” could be correct.

3.      Results from multiple regression models could be reported and conclusions could be based on those results.

NA

Author Response

REVIWER #1

  1. The author claimed conducting a two-stage cluster analysis, but it is not clear from the statistical methods, how this analysis was conducted.

In Data Analysis, we added sentences describing details of two-step (not two-stage) cluster analysis as follows.

After calculating mean, SD, skewness, and kurtosis of all of the variables, we correlated scores of unwanted pregnancy and unhappy pregnancy. A two-step cluster analysis was performed using the scores of unintended and unhappy pregnancies with log likelihood method and Bayesian Information Criterion (BIC) to define the number of clusters. Cluster analysis identifies types according to individual differences [43]. There are a few techniques of cluster analysis. Agglomerative hierarchical cluster analysis is widely used but is characterised by ambiguity of determining the appropriate number of clusters. K-means cluster analysis [44] demands the researcher to determine the number of clusters a priori. We used two-step cluster analysis in this study. The procedure starts with the construction of a cluster features tree that creates ‘nodes’ containing multiple cases. In the second step, agglomerative clustering is used to produce a range of solutions. It automatically confirms the possible maximum number of clusters. This will be followed by determination of the best cluster model in terms of the highest distance increase (measured by Schwarz’s Bayesian Criterion [45] or Akaike Information Criterion [46]) between the two closest cluster models during each stage of the hierarchical clustering (Sarstedt, & Mooi, 2014; SPSS, 2001). Two-step cluster analysis can also deal with large data files efficiently. Two-step cluster analysis has been used in many psychosocial studies (e.g., Carbone et al., 2019; Forsman et al., 2019; Hörz-Sagstetter et al., 2020). As compared with the other methods that identify groups (clusters) of subjects including latent class analysis, the performance of two-step analysis is equally excellent. Silhouette cofficient analysis was taken to evaluate the silhouette of clusters [47, 48]. The clusters were compared in terms of the scores of unintended and unhappy pregnancies. To determine the construct validity of the obtained clusters, we compared the clusters’ mean scores of demographic and obstetrics features, borderline personality traits, adult attachment style, perceived impact of pregnancy, depression, fear of childbirth, obsessive-compulsive symptoms, and foetal bonding disorder.

  1. Table 1, results show that the characterisation may not have been performed properly. Frequencies in the upper left side of table 1 did not convince me that the number (n = 139) for “Cluster 2: unwanted or unhappy” could be correct.

We understand the reviewer’s concern. Therefore, we added the following sentences in Results.

By spotting each case of the three clusters in Table 1, it was found that the Clutser 1 (Wanted and Happy) cases were located in the right two cells of the ‘very pleased’ row whereas the Cluster 3 (Unwanted but happy) cases were located in the left three cell of the ‘very pleased’ row. The Cluster 2 (Unwanted and Unhappy) cases were located in the reaining cells.

  1. Results from multiple regression models could be reported and conclusions could be based on those results.

We are not very clear about the purpose of use of multiple regression analysis here. What does the reviewer recommend us to use as the criterion (dependent) variable and what as the explanatory (independent) variables? In such a case how would the reviewer recommend to avoids the biases due to multiple comparisons?

Reviewer 2 Report

The paper is ambiguous in relation to the few selection criteria and the results presented. Therefore, I suggest that the authors take into account the following considerations

Abstract: It should be reformulated once the following suggestions have been taken into account

INTRODUCTION: The authors define "psychological maladjustments" and psychological disorders as the same entity (Citations 1-5). However, they are not the same and later mention tocophobia as a disorder (6-7) and it is not included as a diagnosis in the DSM5-TR.

In line 56 the authors cite Bustman MN et al who reported neonatal deaths within 27 days of birth and in fact the article mentions that 83 (0.9%) newborns died within the first 28 days of birth.

Line 60-61 The authors refer to a study describing an immigrant population, in a vulnerable condition and also with postpartum depression, and this study is focused on the first trimester of pregnancy, which makes it irrelevant and a little inclined to what the authors want to publish.

Finally, in lines 94-95, the authors mention the possible consequences, including obsessive-compulsive disorders. However, in the introduction they do not mention this disorder as a background and during the presentation of the results they do not mention this disorder as an evaluation and in the conclusions they omit it. Therefore, the magnitude of the study is not clearly understood.

Methods: 

Line 100. The authors mention that the study was conducted in women 12 to 15 weeks pregnant, however the abstract mentions that the interviews were conducted in 696 women in their first trimester of pregnancy.

Lines 103-104 The authors mention that there were no exclusion criteria. However, in a study of behavior and mental health, the age of a pregnant woman and her marital status have a considerable influence in relation to some behaviors that may occur at the beginning of pregnancy. 

On the other hand, the authors comment in the methodology that they carried out 2 measurements, however, they only show the results of the first interview, which I consider irrelevant if they mention this second interview or if they include the results and analysis of both measurements.

Meassurements

to measure whether a pregnancy was unwanted or unplanned refer to a 5-point scale, however it is not mentioned which one it was or if it was created by the authors themselves, how it was validated.

Lines 137-143. The authors use a scale that only predicts depression through the identification of symptoms, but this scale does not make the diagnosis, so the authors should note that it only identifies signs of depression, but does not establish the diagnosis.

RESULTS

During the results the authors show no evidence that establishes a relationship between the psychological state of the pregnant women and the different clusters.

At the end of the results, the authors mention that they did analyze the second interview, but they do not show the results, they only say that they were "Similar" and later in the discussion they do not address them. But they do mention at the end of the discussion that there should be a more rigorous follow-up to evaluate the mental state of pregnant women.

Conclusions.

At the end of the abstract the authors establish a "Risk" which they do not measure among women with unwanted and unplanned pregnancy and psychological adjustment and need specific perinatal health assessment and care. However, they no calculated risk factor with any statistical test.

Finally, In the conclusion the authors note that women who have an unwanted and unplanned pregnancy should be target of specialized intensive mental health care during pregnancy. However this study was focused only in first trimester. 

Author Response

REVIWER #2

Abstract: It should be reformulated once the following suggestions have been taken into account

INTRODUCTION: The authors define “psychological maladjustments” and psychological disorders as the same entity (Citations 1-5). However, they are not the same and later mention tokophobia as a disorder (6-7) and it is not included as a diagnosis in the DSM5-TR.

Unlike the 2nd reviewer, we do define “psychological maladjustments” and “psychological disorders” as the same entity. This notion is based on evidence that most categories of mental disorders are not categories (taxon) but dimensions. We list references of taxometric studies of depression most of which indicate that there is no such category as depression (reference see below).

Tokophobia is also known as fear of childbirth. It has been recognised for a long time as a major perinatal mental health issue no matter how it is termed. Tokophobia does not appear in DSM-5 (To the best of our knowledge, DSM-5 has not yet been revised as DSM5-TR). There are some clinical entities that were included in previous versions of the DSM but discarded from the list of disorders whereas some other clinical entities that did not appear in previous versions but appear in later ones. Therefore, we believe that whether the American Psychiatric Association lists tokophobia as a DSM terminology. New findings of empirical research may change the DSM but not vice versa.

In line 56 the authors cite Bustman MN et al who reported neonatal deaths within 27 days of birth and in fact the article mentions that 83 (0.9%) newborns died within the first 28 days of birth.

Thank you for your comments. We revised the text accordingly.

Line 60-61 The authors refer to a study describing an immigrant population, in a vulnerable condition and also with postpartum depression, and this study is focused on the first trimester of pregnancy, which makes it irrelevant and a little inclined to what the authors want to publish.

Christensen et al. (2011) studied low-income Hispanic immigrants (n = 215) five times from early pregnancy to 12-months postpartum. They found the depression trajectory pattern was linked to unintendedness of pregnancy that was rated at 18 weeks gestation. Our study collected pregnant women at 12 to 15 weeks’ gestational age. Does such a difference of gestation age matter?

Finally, in lines 94-95, the authors mention the possible consequences, including obsessive-compulsive disorders. However, in the introduction they do not mention this disorder as a background and during the presentation of the results they do not mention this disorder as an evaluation and in the conclusions they omit it. Therefore, the magnitude of the study is not clearly understood.

In Introduction, we mentioned about OCD.

Methods:

Line 100. The authors mention that the study was conducted in women 12 to 15 weeks pregnant, however the abstract mentions that the interviews were conducted in 696 women in their first trimester of pregnancy.

We dot think that that is a problem.

Lines 103-104 The authors mention that there were no exclusion criteria. However, in a study of behavior and mental health, the age of a pregnant woman and her marital status have a considerable influence in relation to some behaviors that may occur at the beginning of pregnancy.

We cannot understand what the reviewer suggested us to do.

On the other hand, the authors comment in the methodology that they carried out 2 measurements, however, they only show the results of the first interview, which I consider irrelevant if they mention this second interview or if they include the results and analysis of both measurements.

The data of Wave 2 are relevant. Please read the final paragraph of Results.

Measurements

to measure whether a pregnancy was unwanted or unplanned refer to a 5-point scale, however it is not mentioned which one it was or if it was created by the authors themselves, how it was validated.

In Measurement, we clearly noted we created as hoc items. We cannot understand such simple question items should be ‘validated’.

Lines 137-143. The authors use a scale that only predicts depression through the identification of symptoms, but this scale does not make the diagnosis, so the authors should note that it only identifies signs of depression, but does not establish the diagnosis.

We used the two items of MDE as a ‘flag’ of diagnosis of MDE. We cited the previous research works to justify this assumption (see Measurement).

RESULTS

During the results the authors show no evidence that establishes a relationship between the psychological state of the pregnant women and the different clusters.

We think we did it.

At the end of the results, the authors mention that they did analyze the second interview, but they do not show the results, they only say that they were “Similar” and later in the discussion they do not address them. But they do mention at the end of the discussion that there should be a more rigorous follow-up to evaluate the mental state of pregnant women.

We cannot understand the points of the reviewer’s critiques.

Conclusions.

At the end of the abstract the authors establish a "Risk" which they do not measure among women with unwanted and unplanned pregnancy and psychological adjustment and need specific perinatal health assessment and care. However, they no calculated risk factor with any statistical test.

We cannot understand why we should not use the term ‘risk’ at the palaces the reviewer mentioned to.

Finally, In the conclusion the authors note that women who have an unwanted and unplanned pregnancy should be target of specialized intensive mental health care during pregnancy. However this study was focused only in first trimester.

Does the reviewer note that women who have an unwanted and unplanned pregnancy should be target of specialized intensive mental health care during pregnancy? If so, we cannot follow the logics of the reviewer. Does the reviewer say we does not have to initiate mental health service to expectant women who have an unwanted and unplanned pregnancy.

List of taxometric studies on depression

Ahmed, A. O., Green, B. A., Clark, C. B., Stahl, K. C., & McFarland, M. E. (2011). Latent structure of unipolar and bipolar mood symptoms. Bipolar Disorders, 13, 522-536.

Ambrosini, P. J., Bennett, D. S., Cleland, C. M., & Haslam, N. (2003). Taxonicity of adolescent melancholia: A categorical or dimensional construct? Journal of Psychiatric Research, 36, 247-256.

Balbuena, L., Baetz, M., & Bowen, R. C. (2015). The dimensional structure of cycling mood disorders. Psychiatry Research, 228, 285-294.

Baldwin, G., & Shean, G. (2006). A taxometric study of the Center for Epidemiological Studies Depression Scale. Genetic, Social, and General Psychology Monographs, 132(2), 101-128.

Baptista, M. N., Cunha, F., & Hauck, N. (2019). The latent structure of depression symptoms and suicidal thoughts in Brazilian youths. Journal of Affective Disorders, 254, 90-97.

Franklin, C. L., Strong, D. R., & Greene, R. L. (2002). A taxometric analysis of the MMPI-2 depression scales. Journal of Personality Assessment, 79(1), 110-121.

Gibb, B. E., Alloy, L. B., Abramson, L. Y., Beevers, C. G., & Miller, I. W. (2004). Cognitive vulnerability to depression: A taxometric analysis. Journal of Abnormal Psychology, 113(1), 81-89.

Guo, F., Chen, Z., & Ren, F. (2014). The latent structure of depression among Chinese: A taxometric analysis in a nationwide urban sample. PsyCh Journal, 3, 234-244.

Hankin, B. L., Fraley, R. C., Lahey, B. B., & Waldman, I. D. (2005). Is depression best viewed as a continuum or discrete category? A taxometric analysis of childhood and adolescent depression in a population-based sample. Journal of Abnormal Psychology, 114(1), 96-110.

Holland, J. M., Neimeyer, R. A., Boelen, P. A., & Prigerson, H. G. (2009). The underlying structure of grief: A taxometric investigation of prolonged and normal reaction to loss. Journal of Psychopathology and Behavioral Assessment, 31, 190-201.

Holland, J. M., Schutte, K. K., Brennan, P. L., & Moors, R. H. (2010). The structure of late-life depressive symptoms across a 20-year span: A taxometric investigation. Psychology and Aging, 25(1), 142-156.

Liu, R. T. (2016). Taxometric evidence of a dimensional latent structure for depression in an epidemiological sample of children and adolescents. Psychological Medicine, 46(6), 1265-1275.

Liu, R. T., Burke, T. A., Abramson, L. Y., & Alloy, L. B. (2018). The behavioral approach system (BAS) model of vulnerability to bipolar disorder: Evidence of a continuum in BAS sensitivity across adolescence. Journal of Abnormal Child Psychology, 46(6), 1333-1349.

Liu, R. T., McArthur, B. A., Burke, T. A., Hamilton, J. L., Giollabhul, N. M., Stange, J. P., Hamlat, E. J., Abramson, L. Y., & Alloy, L. B. (2019). A latent structure analysis of cognitive vulnerability to depression in adolescence. Behavior Therapy, 50(4), 755-764.

Okumura, Y., Sakamoto, S., & Ono, Y. (2009a). Latent structure of depression in a Japanese sample: Taxometric procedures. Australian and New Zealand Journal of Psychiatry, 43(7), 666-673.

Okumura, Y., Sakamoto, S., Tomoda, A., & Kijima, N. (2009b). Latent structure of self-reported depression in undergardates: Using taxometric procedures and information-theoretic latent variable modelling. Personality and Individual Differences, 46, 166-171.

Prisciandaro, J. J., & Roberts, J. E. (2005). A taxometric investigation of unipolar depression in the National Comorbidity Survey. Journal of Abnormal Psychology, 114(4), 718-728.

Prisciandaro, J. J., & Roberts, J. E. (2011). Evidence for the continuous latent structure of mania in the Epidemiologic Catchment Area from multiple latent structure and construct validation methodology. Psychological Medicine, 41, 575-588.

Richey, J. A., Schmidt, N. B., Lonigan, C. J., Phillips, B. M., Catanzaro, S. J., Laurent, J., Gerhardstein, R. R., & Kotov, R. (2009). The latent structure of child depression: A taxometric analysis. Journal of Child Psychology and Psychiatry, 50(9), 1147-1155.

Ruscio, A. M., & Ruscio, J. (2002). The latent structure of analogue depression: Should the Beck Depression Inventory be used to classify groups? Psychological assessment, 14(2), 135-145.

Ruscio, J., Brown, T. A., & Ruscio, A. M. (2009). A taxometric investigation of DSM-IV major depression in a large outpatient sample. Assessment, 16(2), 127-144.

Ruscio, J., & Ruscio, A. M. (2000). Informing the continuity controversy: A taxometric analysis of depression. Journal of Abnormal Psychology, 109(3), 473-487.

Ruscio, J., Zimmerman, M., McGlinchey, J., Chelminski, I., & Young, D. (2007). Diagnosing major depressive disorder XI: A taxometric investigation of the structure underlying DSM-IV symptoms. Journal of Nervous and Mental Disease, 195(1), 10-19.

Schmidt, N. B., Kotov, R., Bernstein, A., Zvolensky, M., J., Joiner, T. F. Jr., & Lewinsohn, P. M. (2007). Mixed anxiety depression: Taxometric exploration of the validity of a diagnostic category in youth. Journal of Affective Disorders, 98, 83-89.

Shean, G. D., & Baldwin, G. (2012). The latent structure of the Center for Epidemiological Studies-Depression Scale. Journal of Psychopathology and Behavioral Assessment, 34, 502-509.

Slade, T., & Andrews, G. (2004). Latent structure of depression in a community sample: A taxometric analysis. Psychological Medicine, 35, 489-497.

Solomon, A., Ruscio, J., Seeley, J. R., & Lewinsohn, P. M. (2006). A taxometric investigation of unipolar depression in a large community sample. Psychological Medicine, 36, 973-985.

Strong, D. R., Brown, R. A., Kahler, C. W., Lloyd-E. E., & Niaura, R. (2004). Depression proneness in treatment-seeking smokers: A taxometric analysis. Personality and Individual Differences, 36, 1155-1170.

Whisman, M. A., & Pinto, A. (1997). Hopelessness depression in depressed inpatient adolescents. Cognitive Therapy and Research, 21(3), 345-358.

Authors and year

Measurements

Statistics

N

Dim vs. categ

Ahmed et al., 2011

CIDI

MAMBAC, MAXEIG, L-MODE

20,013

C

Ambrosini et al., 2003

K-SADS, BDI (melancholia)

MAXCOV, MAMBAC

378

C

Baldwin & Shean, 2006

CES-D, DIS

MAMBAC, MAXEIG

392

D

Balbuena et al., 2015

Affective Lability Scale-Short Form, Mood Disorders Questionnaire, N of EPQ

MAMBAC, MAXEIG, L-MODE

319

D

Baptista et al., 2019

MDE

MAMBAC, MAXEIG, L-MODE

2,587

D

Franklin et al., 2002

MMPI-2

MAXCOV, MAMBAC

2,000

D

Gibb et al., 2004

Cognitive Style Questionnaire, Dysfunctional Attitude Scale

MAMBAC, MAXEIG, L-MODE

2,114

D

Guo et al., 2014

CESD

MAMBAC, MAXEIG, L-MODE

6,132

D

Hankin et al., 2005

Child and Adolescent Psychopathology Scale

MAXCOV

845

D

Holland et al., 2009

Inventory of Complicated Grief-Revised

MAMBAC, MAXEIG

1,069

D

Holland et al., 2010

Health and Daily Living Scale (MDE)

MAXEIG

1,289

D

Liu, 2016

MDE

MAMBAC, MAXEIG, L-MODE

1,288

D

Liu et al., 2018

BIS/BAS Scale

MAMBAC, MAXEIG, L-MODE

12,494

D

Liu et al., 2019

negative inferential style, ruminative response style, self-referent information processing

MAMBAC, MAXEIG, L-MODE

485

D

Okumura et al., 2009a

CESD

MAMBAC, MAXEIG

20,987

D

Okumura et al., 2009b

SDS

MAMBAC, MAXEIG

2,187

D

Prisciandaro & Roberts, 2005

CIDI

MAMBAC, MAXEIG

4,577

D

Prisciandaro & Roberts, 2011

Manic Episode symptoms

MAXCOV, MAMBAC

10,105

D

Richey et al., 2009

CDI

MAXCOV, MAMBAC, MAXEIG, L-MODE, MAXSLOPE

1,531

C

Ruscio & Ruscio, 2000

BDI/SDS + MMPI

MAXCOV, MAMBAC

34,960

D

Ruscio & Ruscio, 2002

BDI

MAMBAC, MAXEIG, L-MODE

2,260

D

Ruscio et al. 2007

MDE

MAXCOV, MAMBAC, MAXEIG

1,800

C

Ruscio et al., 2009

ADIS-IV-L (MDE)

MAXEIG, MAMBAC

1,500

C

Schmidt et al., 2007

Youth Self Report, CBCL, K-SADS

MAXCOV, MAMBAC

1,709

C

Shean & Baldwin, 2012

CESD

MAXEIG, MAMBAC

3,395

D

Slade & Andrews, 2004

CIDI (MDE), K10

MAXCOV, MAMBAC

1,933

D

Solomon et al., 2006

BDI

MAXEIG

1,400

C

Strong et al., 2004

Depression Proneness Inventory

MAMBAC, MAXEIG, L-MODE

439

C

Whisman & Pinto, 1997

Hopelessness Scale for Children, BDI

MAMBAC

160

D